# The implementation of NEMS GFS Aerosol Component (NGAC) Version 2.0 for global multispecies forecasting at NOAA/NCEP: Part I Model Descriptions

Jun Wang[1], Partha S. Bhattacharjee[2], Vijay Tallapragada[1], Cheng-Hsuan Lu[3], Shobha Kondragunta[4], Arlindo da Silva[5], Xiaoyang Zhang[6], Sheng-Po Chen[3], Shih-Wei Wei[3], Anton S. Darmenov[5], Jeff McQueen[1], Pius Lee[7], Prabhat Koner[8], Andy Hurris[8]

[1]NOAA/NWS National Centers for Environment Prediction, College Park, 20740, USA
[2]I. M. Systems Group, INC. at NOAA/NWS National Centers for Environment Prediction, College Park, 20740, USA
[3]University at Albany, State University of New York, Albany, 12222, USA
[4]NOAA/NESDIS Satellite Applications and Research Center, College Park, 20740, USA
[5]NASA Goddard Space Flight Center, Greenbelt, 20771, USA
[6]South Dakota State University, Brookings, 57007, USA
[7]NOAA/OAR Air Resource Laboratory, College Park, 20740, USA
[8]University of Maryland, Earth System Science Interdisciplinary Center, College Park, 20740, USA

*Correspondence to*: Jun Wang (jun.wang@noaa.gov)

**Abstract.** The NEMS GFS Aerosol component version 2.0 (NGACv2) for global multi-species aerosol forecast has been developed at the National Centers of Environment Prediction (NCEP) in collaboration with the NESDIS Center for Satellite Applications and Research (STAR), NASA Goddard Space Flight Center (GSFC), and University at Albany, State University of New York (SUNYA). This paper describes the continuous development of the NGAC system at NCEP after the initial global dust-only forecast implementation (NGAC version 1.0, NGACv1). With NGACv2, additional sea salt, sulfate, organic carbon and black carbon aerosol species were included. The smoke emissions are from the NESDIS STAR's Global Biomass Burning Product (GBBEPx), blended from the global biomass burning emission product from a constellation of geostationary satellites (GBBEP-Geo) and GSFC's Quick Fire Emission Data Version 2 from a polar orbiting sensor (QFED2). This implementation advanced the global aerosol forecast capability and made a step forward toward developing a global aerosol data assimilation system. The aerosol products from this system have been used by many applications such as for regional air quality model lateral boundary conditions, satellite SST physical retrievals and the global solar insolation estimation. Positive impacts have been seen in these applications.

## 1 Introduction

Aerosols affect the atmospheric energy budget by scattering and absorbing solar and thermal radiation, and by interacting with clouds. The impact of aerosols on the radiation interaction processes varies with different aerosol species. It is known

that sulfate aerosols predominantly reflect sunlight and cool the atmosphere, while black carbon aerosols absorb radiation and warm the atmosphere (Haywood and Boucher, 2001). Organic carbon aerosols also warm the atmosphere depending on the brightness of the underlying ground. Dust impacts radiation to varying degrees depending on the composition of the minerals in the dust grains, and whether they are coated with black or brown carbon (Sokolik and Toon-1999). Sea salt

particles scatter the incoming solar radiation and absorb the outgoing terrestrial radiation, with short and long wave radiation approximately the same order of magnitude, but in opposite sign (Lundgren, 2013). In addition to the effect on the atmospheric energy budget, the composition and size distribution of aerosols impact their effectiveness as cloud condensation nuclei (CCN) and result in variations of the distribution of CCN (Mircea, et al. 2002). The change in the cloud properties further impacts cloud albedo, cloud lifetime, precipitation and vertical atmospheric heating profile, etc. (Twomey,

1977; Albrecht, 1989; Lohmann and Feichter, 2005; Stevens and Feingold, 2009; Rosenfeld et al. 2014). To represent the diverse aerosol properties and estimate their effects on physical radiation and cloud processes, typical aerosol size distributions are adopted. The effect of the physical processes involving aerosols are not limited to climate studies but also effect other earth science systems. Polluted air with an increased amount of aerosols tends to generate bright clouds reducing precipitation efficiently, which then leads to a weak regional hydrological cycle that affects the quality of fresh water over

the tropics and the subtropics, especially in the Asian region which has the large tropical and subtropical aerosol emission sources (Ramanathan et al. 2001). Minerals such as nitrogen, phosphorus and iron that are deposited on land and oceans due to aerosol landing may stimulate productivity in some land ecosystems (e.g., tropical forecast) and marine ecosystems, enhancing $CO_2$ intake and the biogeochemical cycles (Jickells et al. 2005; Mahowald, 2011).

Aerosol impact on weather prediction has been investigated extensively in recent years. Many studies show that aerosols

may have significant impact on severe weather events. Rosenfeld (2012) indicated that microphysical and thermodynamic effects from aerosols have significant impact on tropical cyclone development. Wang (2014) showed that anthropogenic aerosols from Asian pollution increased the precipitation and poleward heat transport, thereby intensifying the Pacific storm track. Saide (2015) analyzed historical tornado outbreak data and concluded that an increase in aerosols can induce tornado outbreaks when atmospheric conditions are favorable for severe thunderstorm development. Fan (2015) also showed that

anthropogenic aerosols contributed to catastrophic floods in Southwest China in 2013. These studies illustrate the importance of including a more realistic treatment of aerosol–cloud interactions in numerical weather prediction (NWP) models.

Many major NWP operational centers around the world have started to investigate the impact of aerosol on medium range global weather forecasting. Tomkins et al. (2005) showed that an updated dust climatology leads to a northward shift of the African Easterly Jet (AEJ) in the European Centre for Medium-Range Weather Forecasts (ECMWF) NWP model, which

agrees with the observations. Their study confirmed that a better representation of the seasonal distribution of aerosol (especially dust) improves the model mean state and local surface weather forecast skill. Reale et al. (2011) studied the impact of aerosol on global weather forecast skill using NASA's Goddard Earth Observing System version 5 (GEOS-5) and confirmed that forecasts with interactive aerosol radiation effects predicted a more realistic thermal structure and AEJ location in the African monsoon region. They suggested designing an event–focused system to activate aerosol radiation

interaction in a global forecast model when there is a strong aerosol event. Grell and Baklanov (2011) suggested that a fully coupled chemical and weather forecast model should be used for weather forecasts and air quality predictions due to the positive improvement observed in temperature and wind forecasts during wild fire events. Therefore, in order to achieve better forecast performance, comprehensive representations of aerosol direct and indirect effects and aerosol-aware physics schemes are required in the high resolution weather forecast models. Murphy (2014) investigated aerosol complexity in the global NWP configuration of the Met Office Unified Model (MetUM). They concluded that aerosol species treated as prognostic variables help to predict aerosol events, and when the direct and indirect aerosol effects are represented in the model the radiation bias is reduced and the regional temperature and height forecast is improved for the aerosol events. Zhang (2016) investigated the changes in solar radiation forcing from a smoke event and the corresponding changes in surface cooling and model bias. However, they found that the inclusion of realistic smoke aerosol fields in the forecast model itself is not sufficient to get significant improvement in surface temperature forecasts considering the current range for model temperature uncertainty.

A unified modelling framework for both weather forecast and climate prediction is under development at the National Centers for Environmental Prediction (NCEP). Specifically, NCEP has been developing NOAA Environmental Modeling System (NEMS) as its next-generation operational system (Black et al., 2007, 2009) and has collaborated with NASA/Goddard Space Flight Center (GSFC) to develop NEMS GFS Aerosol Component (NGAC) for predicting the distribution of atmospheric aerosols (Lu et al., 2010). Implemented in 2012, NGAC version 1 provided the first operational global dust aerosol forecasting capability at NCEP (Lu et al. 2016). It used an in-line aerosol module based on the Goddard Chemistry Aerosol Radiation and Transport (GOCART) model within GEOS-5. The system was built upon the Earth System Modeling Framework (ESMF), which provided the techniques to implement exchangeable and reusable earth science system components. The atmosphere model was equivalent to the 2010 operational GFS, but running at lower resolution (T126, approximately 100 km). The model provided a five-day dust only forecast globally at 1º ×1º resolution once per day at 00 Coordinated Universal Time (UTC).

Based on NGAC version 1, NCEP implemented a multi-species aerosol forecast capability through continuous collaboration among NCEP, NASA/GSFC, NESDIS Center for Satellite Applications and Research (STAR) and University at Albany, State University of New York (UAlbany). In Section 2, we describe the NGACv2 model configuration. In Section 3, we present the operational implementation of NGACv2. In Section 4, we present some results of NGACv2 forecasts. In Section 5, we demonstrate three examples of NGACv2 downstream applications. Section 6 provides concluding remarks. Detailed NGACv2 verification will be presented in a companion paper (Bhattacharjee et al. 2018)).

## 2 Model descriptions

NGAC is an interactive atmospheric aerosol forecast system with the NEMS global spectral model (NEMS GSM) as the atmosphere model and GOCART as the aerosol model. The system was built upon the ESMF infrastructure to streamline the

sub-components in the earth system. NEMS GSM has been developed at NCEP to implement the standalone global forecast system (GFS) in the NEMS framework since 2006 (Black et al., 2007, 2009). NGAC shares the same global forecast system with NEMS GSM. Detailed information on the NEMS GSM can be found in NGACv1 (Lu et al. 2016). The major model updates from NGACv1 to NGACv2 are listed below.

## 2.1 Updates in NEMS GSM

The physics package in NEMS GSM has been updated to the operational GFS physics package for each GFS implementation until NEMS GSM was implemented into operation in 2017. There have been several important physics updates in NEMS GSM since NGACv1 was implemented in 2012. During NGACv2 development, the GFS physics package was implemented into operations in January 2015 with slightly improved performance and this version of physics package was implemented

into NEMS GSM in October 2015. NGACv2 uses the same physics package as the 2015 version of the operational GFS. Major GFS physics updates since 2012 are listed below.

First, the radiation package Rapid Radiative Transfer Model (RRTM) was upgraded to the Monte Carlo Independent Column Approximation (McICA) radiation package (Pincus et al. 2003). The new radiation package can address sub-grid cloud variability; in particular it can be applied to the situation with vertically overlapping fractional clouds and when the cloud

condensates form inhomogeneously. The planetary boundary layer (PBL) scheme is updated to the hybrid Eddy-Diffusivity Mass-Flux (EDMF) scheme (Han et al. 2015).

Besides the updates in the radiation package and PBL scheme, changes are also made in the land surface scheme. The prescribed soil moisture climatology used in the soil moisture nudging scheme is upgraded from CPC's bucket soil moisture climatology to the climatology derived from NCEP Climate Forecast System (CFS) and Global Land Data Assimilation

(GLDAS). To enhance the weak land–atmosphere coupling strength in the GFS, the ratio of thermal to momentum roughness is modified as a function of vegetation type. A look-up table based on the vegetation type replaced the 1.0 degree momentum roughness length climatology to better describe the roughness length. As in NGACv1 (Lu, et al. 2016), the NGACv2 uses the Relaxed Arakawa–Schubert scheme with enhanced tracer treatment (the RAS scheme, Moorthi and Suarez, 1992, 1999) which provides the convective mass fluxes at each model layer in the cloud for vertical aerosol

transport. The NEMS GSM uses the Simplified Arakawa–Schubert scheme (the revised SAS scheme, Han and Pan, 2011) where these convective mass fluxes are currently not available.

## 2.2 Aerosol model

In 2012, NCEP implemented NASA/GSFC's GOCART aerosol module (Colarco et al. 2010) into NGACv2 and NCEP operations. The GOCART module in NGACv1 can simulate atmospheric aerosols (including sulfate, black carbon (BC),

organic carbon (OC), dust, and sea-salt), and sulfur gases (Chin et al. 2000, 2002, 2003, 2004, 2007, 2009; Ginoux et al. 2001, 2004; Bian et al. 2010; Colarco et al. 2010; Kim et al. 2013), but only the dust module is turned on in NGACv1. In NGACv2 the GOCART module is updated and the suite of aerosol components is turned on to predict a wider range of

aerosols. Figure 1 shows the summary of the in-line GOCART aerosol module that is used in GOES-5 and NGACv2. Details of aerosol loss processes for all components including dry deposition, wet removal, and convective scavenging processes are specified in Chin et al. (2002). In NGACv2, a computational error on dust AOD calculation is fixed, and the removal process has been tuned to improve model performance. Black carbon and organic carbon aerosols are tracked separately in

GOCART. The organic carbon is presented as particulate organic matter. The chemical processing of carbonaceous aerosols as a conversion from a hydrophobic to hydrophilic mode follows Cooke et al. (1999) and Chin et al. (2002) with an e-folding timescale of 2.5 days (Maria et al. 2004). Following Colarco (2014), five size bins of sea salt aerosol particles with a dry radius range of 0.03-10mm are considered for an indirect production mechanism from bursting bubbles (Monahan et al. 1986), as modified by Gong (2003). Four sulfate tracers, i.e. dimethyl sulfide (DMS), $SO_2$, $SO_4$, and methane sulfonic acid

(MSA) are tracked. Sulfate chemistry includes the DMS oxidation by OH during the day and by $NO_3$ at night to form $SO_2$, and $SO_2$ oxidation by OH in the gas phase and by $H_2O_2$ in the aqueous phase to form sulfate, as described in Chin et al. (2002). The aerosol optical thickness (AOT) is computed from the complex refractive indices, size distributions, and the hygroscopic properties of aerosols following Chin et al. (2002).

## 2.3 Emissions

With the exception of biomass burning emissions, NGACv2 adopts GEOS-5 GOCART aerosol emissions. Emissions of carbonaceous aerosols and $SO_2$ from biomass burning are obtained from Global Biomass Burning Emission Product-extended (GBBEPx) described in Zhang et al. (2014). GBBEPx emissions are blended from NESDIS's Global Biomass Burning Emission Product from a constellation of geostationary satellites (GBBEP, Zhang et al. 2012) and GMAO's Quick Fire Emissions Data Version 2 from polar orbiting satellites (QFED2, Darmenov and da Silva, 2015). GBBEPx provides

global emissions for $CO_2$, CO, OC, BC, $PM_{2.5}$, and $SO_2$ daily. The operational implementation of GBBEPx product at NESDIS enables NCEP to upgrade NGAC from dust-only to multi-species aerosol forecasts (including dust, sea salt, sulfate, and carbonaceous aerosols).

Table 1 summarizes the emissions for different aerosol types used by the GOCART aerosol module in NGACv2. Emissions datasets are re-gridded to the native model grid (i.e., T126 Gaussian grid, about 100km horizontal resolution). The emission

sources/algorithms used for each species are as follows:

- For sulfate aerosols, primary emissions of DMS, $SO_2$, and $SO_4$ are considered. Daily biomass burning emissions are taken from NESDIS GBBEPx dataset described above. Anthropogenic emissions of $SO_2$ are taken from the Emissions Database for Global Atmospheric Research (EDGAR), version 4.1 (European Commissions, 2010). For anthropogenic emissions of primary sulfate, the AeroCom Phase II dataset (HCA0 v1, Diehl et al., 2012) is used. $SO_2$ emissions from

30        ocean-going ships are taken from EDGAR v4.1, and ship $SO_4$ emissions, taken from AeroCom Phase II (HCA0 v1), are derived from gridded emissions data set of Eyring et al. (2005) using the EDGAR v4.1 $SO_2$ ship emissions. Aircraft emissions of $SO_2$ are derived from the AeroCom Phase II (HCA0 v1), which in turn is based on NASA's Atmospheric Effects of Aviation Program (AEAP) inventory. DMS emissions from marine algae are calculated from DMS

concentrations and water-to-air transfer velocity (piston velocity). Monthly-varying DMS concentrations are taken from Lana et al. (2011). Piston velocity is computed from 2-m temperature and 10-meter wind following the empirical formula from Liss and Merlivat (1986).

- The sources for carbonaceous aerosols arise from anthropogenic and natural sources, including biomass burning, fossil fuel, biofuel, and (in the case of OC) from the oxidation of biogenic emissions. Biomass burning sources are taken from daily GBBEPx dataset described above. For anthropogenic emissions, AeroCom Phase II data set (HCA0 v1) is used. This data set is based on gridded inventory from Bond et al. (2004) and yearly global emission trends compiled from Streets et al. (2008, 2009). Ship emissions are determined in the same way as for SO4 (AeroCom Phase II, HCA0 v1). Emissions of terpene from vegetation are oxidized to produce OC aerosols. Biogenic emissions are treated following Chin et al. (2002) using a monthly varying Global Emissions Inventories Activity (GEIA) inventory (Guenther et al. 1995).

- For natural aerosols, the emissions are largely driven by variability in model dynamics (specifically, the surface wind). Dust emissions use a map of potential dust source locations based on topographic depressions (Ginoux et al., 2001). The uplifting of dust aerosols depend on the wind speed formulation of Marticorena and Bergametti (1995). The parameterization of sea salt emissions follows the formulation of Gong (2003).

Time frequency of emissions differs between different sources. Both dust and sea salt have wind-speed dependent emissions, updated every time step. Biomass burning emissions from GBBEPx are updated daily. For other emissions, annually- and monthly-varying emissions are temporally interpolated using linear interpolation. For instance, anthropogenic carbonaceous and primary sulfate aerosols emissions from AeroCom Phase II, HCA0 v1 cover the period 1976-2006. For retrospective NGACv2 experiments, emissions are linearly interpolated between the available years. For real-time NGACv2 forecasts, the latest available emissions (2006 emissions) are used.

## 3 NGAC version 2 operational implementation

A phased implementation approach is used for the NGAC implementation at NCEP. The first implementation was for dust-only forecasts. The current NGACv2 implementation documented here enabled the capability of multi-species aerosol forecast including carbonaceous aerosols, sea salt and sulfate aerosols. An aerosols data analysis system capability is under development targeted for the third phase of the NGAC upgrade. The phased implementation also includes both science and software upgrades in the global forecast system.

Effective on March 7, 2017, starting with the 00:00UTC cycle, NCEP began to run and disseminate data from the NGACv2 system operationally. NGACv2 runs at T126 L64 resolution and provides 5-day multi-species aerosols forecasts, twice per day for the 00:00UTC and 12:00UTC cycles. The NGACv2 initial conditions are taken from the 12 hour NGACv2 forecast

from the previous cycle while meteorological initial conditions are from the down-scaled high-resolution Global Data Assimilation System (GDAS) analysis.

As specified in Section 2, the NGACv2 has an atmosphere model updated to the latest GFS, implemented in May 2016. However, NGACv2 uses same configuration as operational GFS except a different convection scheme. NGACv2 provides products in addition to those from NGACv1 dust-related products. First, total Aerosol Optical Depth (AOD) and AOD from each species are produced to support global and regional multi-model ensemble aerosol forecasts. Second, single scattering albedo and asymmetric factor for total aerosols at 340nm are produced to support UV index forecast. Third, three-dimensional mixing ratios for each aerosol species at model levels are produced to support NCEP's operational regional Community Multiscale Air Quality model (CMAQ) and satellite SST retrieval. A complete list of the new output fields is in Appendix A.

## 4 NGAC version 2 results

In this section, the results for the emissions and budgets from operational NGACv2 forecasts and a case study are presented. Detailed NGACv2 performance review is presented d in a separate companion paper (Bhattacharjee et al. 2018).

### 4.1 Budgets

A retrospective NGACv2 run was conducted for the June 2015 to Feb 2017 period. Figure 2 shows the global annual emission, burden, and lifetime (or atmospheric residence time) calculated from NGACv2, relative to other similar global aerosol models, including GEOS-4 (Colarco et al., 2010), the models participating in the AeroCom model intercomparison studies (Textor et al., 2006) and NGACv1 (for the case of dust).

Large differences are found in emissions, burdens, and lifetimes within the AeroCom models, which are primarily related to the differences in the emission parameterizations, the particle sizes, the meteorological fields and model configuration used in the individual models (Textor et al., 2006). The simulated total emissions, annual burden, and lifetime from all the aerosol species in NGACv2 are within the range of the AeroCom models. Compared to NGACv1 (the fourth bar in the upper three plots of the first column), NGACv2 has larger dust emissions due to GFS physics updates (2379 vs. 1980 Tg yr$^{-1}$). In NGACv2, the dust lifetime is longer than NGACv1 (7.45 vs 4.3 days) and the annual burden is about 50% more than NGACv1 (30.6 vs 21.9 Tg), but closer to in-line GOCART in GEOS-4 (30.7 vs 31.6 Tg). These results suggest that dust in NGACv2 is closer to GEOS-4 dust when compared to NGACv1. Sea salt emission is lower than in GEOS-4 (8660 vs 9729 Tg yr$^{-1}$) and its burden lifetime is slightly less than that in GEOS-4. In NGACv2, the emissions of black carbon and organic carbon are larger than in GEOS-4 due to the different biomass burning emissions; however, their burden and lifetime are smaller than the GEOS-4 because of the relatively large removal process. Sulfate emission is slightly smaller than that in GEOS-4 (55.47 vs. 58.73 Tg yr$^{-1}$); NGACv2 sulfate also has less burden and lifetime. It is worth noting that the NGAC

emissions, budget, and lifetime presented here are in the context of the AeroCom model suite. Evaluation of aerosol budget, emissions, and lifetime using observations is beyond the scope of this study.

## 4.2 Case study

Figure 3 shows the total aerosol optical depth simulated by NGACv2 during a smoke event from Canadian wildfires during June 27-July 6, 2015. A strong trough and jet stream dominated the middle of North America, and the wind transported the smoke plume toward the southeast from Canada to the Dakotas, Nebraska and several other states and then reached the Great Lakes region. Elevated AOD associated with the smoke plume has been observed by the space-borne Moderate Resolution Imaging Spectroradiometer (MODIS) sensor and Visible Infrared Imaging Radiometer Suite (VIIRS). The spatial pattern is also predicted by International Cooperative for Aerosol Prediction Multi-Model Ensemble (ICAP-MME, Sessions et al. 2015). The AOD simulated by NGACv2 are consistent with ICAP-MME as well as the observations from MODIS and VIIRS imagery. However, NGACv2 fails to capture the large AOD over the south of the Great Lakes that is shown in the satellite retrievals and the ICAP-MME.

ICAP-MME total AOD products are generated from aerosol forecasts from four well-established aerosol models, including NASA/GSFC, ECMWF, Naval Research Laboratory (NRL), and Japan Meteorological Agency (JMA). Near-real-time satellite based smoke emissions are used by the four ICAP core models, e.g., QFED2 for NASA/GSFC, Fire Locating and Modeling of Burning Emissions (FLAMBE) by NRL, and Global Fire Assimilation System (GFAS) by ECMWF and JMA. In additional, aerosol data assimilation has been utilized by all these models to constrain the modelled AOD errors and bring modelled AODs closer to the satellite observations. Less satisfactory performance in NGAC v2 with respect to ICAP-MME suggest the need for additional model tuning. However, the performance differences cannot be attributed to NGACv2 model deficiency alone. Lynch et al. (2016) reported that model tuning process is equally as significant as data assimilation on the model performance. Sessions et al. (2015) reported that ICAP-MME out performs the participating members, providing a valuable aerosol forecast guidance. Therefore, the results that multi-model ensemble from four well-established models with data assimilation capabilities outperforms a single model without data assimilation is somehow anticipated.

## 5 NGAC version 2 application

The implementation of NGACv2 will provide a full suite of 2-dimensional (2-D) and 3-dimensional (3-D) aerosol products for various downstream applications. Examples of NGACv2 product applications are given here.

## 5.1 Dynamic boundary conditions for regional air quality model

One direct application of NGAC is to provide dynamic boundary conditions for regional air quality models such as Community Multiscale Air Quality (CMAQ) modelling system. The utilization of NGAC for CMAQ with zero-flux divergence outflow and prescribed concentrations for inflow chemical lateral boundary conditions for the dust-associated

aerosol has been operational since February 2016 under the auspices of the National Air Quality Forecasting Capability (NAQFC) (Lee et al. 2017). CMAQ previously used static climatology boundary conditions as lateral boundary conditions, which limited the regional forecast capability when an aerosol event moved into the regional domain from the CMAQ boundary. Figure 4 shows an event on June 10-12, 2015 when smoke from Canada was moving into the United States. The

left side panel is the $PM_{2.5}$ forecast on June 10[th], 11[th] and 12[th] from the CMAQ run using GEOS-Chem model 2006 monthly average values for all the aerosol species at the lateral boundary. The middle panel is the $PM_{2.5}$ forecast from CMAQ during the same period using NGACv2 multi-species aerosols as the lateral boundary condition. $PM_{2.5}$ observations in cycled dots are also shown in both panels to compare CMAQ forecast with observations. The right panel is the difference between the two runs. The figure shows that no smoke was predicted over central Canada and the US in the run using the climatology as

the lateral boundary condition; while the run using NGAC multi-species aerosols as the boundary condition shows a large amount of smoke passing the US-Canadian border and coming across the Great Lakes region. The figure shows that using the NGAC forecast as the CMAQ lateral boundary condition significantly improved the CMAQ forecast.

Figure 5 shows the surface $PM_{2.5}$ with frontal passages in the June 9-12 2015 Canadian fire. Panel a) is the averaged surface $PM_{2.5}$ from 95 sites in the central United States from June 7-15. Panel b) is the averaged surface $PM_{2.5}$ from 82 sites in the

northeast United States during the same time period. The line with circular dots is observations. The black line is the CMAQ forecast with climatology as the lateral boundary condition. The blue line is CMAQ using the operational NGAC dust-only forecast as the lateral boundary condition. The red line is the CMAQ forecast using NGACv2 multi-speces aerosol forecasts. It is clear that the run with the NGACv2 forecast is closer to observations than the runs from the other experiments even though the peak of $PM_{2.5}$ in this run is still lower than the observations.

**5.2 Satellite SST retrieval**

An example presented in Figure 6 focused on exploring and refining the use of aerosol information in physical deterministic retrievals of Sea-Surface Temperature (SST). This experiment has been conducted for night time scenarios using matchup data and cloud-free conditions identified using an experimental filter (EXF, Koner et al. 2016). NGAC 3-D aerosol predictions are used as input to the Community Radiative Transfer Model (CRTM), along with GFS profiles of humidity and

temperature. Aerosol column density (ACD) of all aerosols is represented by the single Jacobian value that are calculated in CRTM representing the derivative of radiation transfer equation with respect to a single variable, the ACD is then included in the state vector for the MODIS-Aqua SST retrieval. Additional channels available for MODIS, combined with a 3-element reduced state vector, offer the prospect of testing a variant of the Truncated Total Least Squares (TTLS, Koner and Harris, 2016) approach. A comparison between results for the 2-component [SST, total column water vapor (TCWV)] for the

Modified Total Least Squares (MTLS, Koner et al. 2015) algorithm and 3-component [SST, TCWV, ACD] state vectors is shown in Figure 6. It can be seen that the RMSE (dashed standard deviation lines) is improved noticeably when ACD is a retrieved parameter. A further consequence of including ACD in the state vector is that algorithm sensitivity is significantly improved. This is demonstrated by the increase in the degree of freedom in retrieval (DFR) values to 0.75 and above. The

NGACv2 aerosol products make it possible to design the TTLS scheme, further development is required to improve the SST retrieval using aerosol products in real operation.

## 5.3 Insolation on the earth surface estimation

An estimation of the insolation on the earth surface is another application where the NGACv2 multi-species aerosol forecast can be directly used. The NGACv2 aerosol forecast AOD products were incorporated to the semi-empirical GOES satellite global and direct horizontal irradiance estimation model (Perez Model) developed at Atmospheric Sciences Research Center (ASRC) in State University of New York at Albany (Perez et al. 1990). The investigation period consisted of three months in the spring of 2016. To compute all sky irradiance, the model needs environmental inputs such as altitude, and atmospheric condition variables such as air temperature, water vapor, AOD, and ozone. Model sensitivity tests were conducted to investigate the relative importance of the factors that influence the insolation on the earth surface. Figure 7a shows sensitivity of the Direct Normal Irradiance (DNI) (Bird et al. 1980) component of the Perez model to factors such as altitude, ozone, moisture, aerosols, and air mass. It can be seen that compared to double ozone, double moisture or a decrease in elevation, double AOD has a larger impact on available DNI. Figure 7b shows aerosol impact on the mean bias error (MBE) of Global Horizontal Irradiance (GHI) from Perez Model. The GHI is computed from Perez Model with AOD and water vapor from different sources. Five sets of experiments were conducted to show the GHI MBE. "GHI" is the control experiment with monthly averaged AOD from NASA Earth Observations (NEO) and water vapor from NEO or GFS. Validation are performed on two sets of sites, one set is validated against GOES eastern US satellite (GOES-13), the other against GOES western US satellite (GOES-15). "GHI_our" experiment is using the monthly averaged AOD from ASRC. "GHI GM" is using Gueymard AOD (Gueymard, 2008), GHI with NGACv2 AOD at 550nm and GHI with NGACv2 AOD at 660nm are also shown. It is found that GHI MBE is the smallest for the experiment using NGAC AOD at 660nm for the 2016 spring period. The results indicate potential improvement in the operational insolation estimate using NGAC AOD at 660nm.

## 6 Conclusions

The implementation of NGACv2 provides operational global multi-species aerosol forecasts at NCEP. Total emissions, annual burden, and lifetime from all the aerosol species in NGACv2 are within the range of the AeroCom models and comparable to those in GEOS-4. More extensive evaluation of NGACv2 is presented in the companion paper by Bhattacharjee et al. (2018). Because the results will be used as baseline for some future development work described below, a general description of the NGACv2 evaluation is shown here. Dust forecast skill is comparable to that in NGACv1 after the bug fix and removal process tuning. The long range dust transport of Sahara dust is slightly improved. Sea salt performs normally compared to other models. Sulfate, black carbon and organic carbon are unrepresentative in North America, the sub-Saharan region and South America in smoke season. Generally fire activity shows up in the right location, but the

concentration is too low compared to observations. The upgrade from dust-only system to multi-species system enables NCEP to produce full suite of aerosol products to serve a wide-range of stakeholders, such as air quality and health professionals, aviation authorities, policy makers, and climate scientists. CMAQ experiments using the NGAC multi-species aerosol forecast as boundary conditions show positive impacts for a smoke event compared to the operational configuration using static boundary conditions. Using NGACv2 AOD 660nm improves the mean bias errors in estimation of surface insolation.

The evaluation of NGACv2 forecast results shows that the initial conditions could have a significant impact on model performance. Currently NGACv2 is using the forecasted aerosol from the previous cycle and downscaled meteorology fields from the NCEP operational high resolution data assimilation analysis. Without real time assimilation the initial aerosol fields may contain errors that propagate to all forecast hours. The aerosol data assimilation using VIIRS in GSI is under development and expected to be implemented in the near future. Future direction for NGAC will be focused on: (1) implementing aerosol data assimilation toward improving aerosol forecasts, and (2) improving the representation of aerosol-radiation-cloud interaction in the atmosphere model toward improving weather forecast and climate prediction.

The National Weather Service (NWS) is transitioning their operational GFS to a unified, fully coupled Next Generation Global Prediction System (NGGPS) within NEMS. The new system will take the most recent advances in weather prediction modeling from NOAA and the research community to extend weather forecasting to 30 days and to improve hurricane track and intensity forecast, in addition to improving medium range weather prediction. This system (Figure 8) is an earth science system with six components including atmosphere, ocean, land, sea ice, wave and aerosol. Recently the NOAA Geophysical Fluid Dynamic Laboratory (GFDL)'s Finite Volume Cubed Sphere (FV3) dynamical core was selected as the new NGGPS atmospheric model. The prototype NGGPS system FV3GFS has been developed by coupling FV3 dynamic core with GFS unified physics suite (the same physics suite in NEM GSM) under the NEMS framework. The first FV3GFS release became public on May 15, 2017. NGAC global aerosol forecast capability is now being transitioned to the FV3GFS system; the NGACv2 forecast performance described above will be used as baseline to evaluate the FV3GFS based aerosol system. Besides the change in the atmosphere model, the development direction on improving aerosol forecast performance, prognostic capability, adding data assimilation component, and coupling with radiation/clouds remains the same.

**Code and data availability**

NCEP Operational Products Suite products are distributed in near real time at NOAA Operational Model Archive and Distribution System (NOMADS).The website is accessible to public users free of charge.  NGACv2 products are available at:

http://www.nomads.ncep.noaa.gov/pub/data/nccf/com/ngac/prod

The source code, scripts, parameters, fixed field files can be obtained at:

http://www.nco.ncep.noaa.gov/pmb/codes/nwprod/ngac.v2.3.0/

Web graphics will remain available at:

http://www.emc.ncep.noaa.gov/gmb/NGAC/html/realtime.ngac.html

NGAC products are encoded in GRIB2. The NCEP grib2 table is updated to include the definition of new aerosol types. Users should download the latest versions of wgrib2 and the other NCEP GRIB utilities to use the NGAC output products.

A website containing retrospective run results from NGACv2 for the period of June 2015-December 2016 is at:

http://www.emc.ncep.noaa.gov/gmb/NGAC/NGACv2/

**Appendix A: New NGAC products**

Output files and the new fields for NGACv2 (Q2FY2017 Implementation)

1. ngac.tCCz.a2dfHHH.grib2, where HHH=00, 03, …, 120 and CC=00, 12:

ASYSFK: Asymmetry Factor at 340 nm from total aerosols [Numeric]

SSALBK: Single Scattering Albedo at 340 nm from total aerosols [Numeric]

AOTK: Aerosol Optical Thickness at 550 nm from total aerosols [Numeric]

AOTK: Aerosol Optical Thickness at 550 nm from sea salt aerosol [Numeric]

AOTK: Aerosol Optical Thickness at 550 nm from black carbon dry aerosol [Numeric]

AOTK: Aerosol Optical Thickness at 550 nm from particulate organic carbon dry aerosol [Numeric]

AOTK: Aerosol Optical Thickness at 550 nm from sulfate dry aerosol [Numeric]

DUST_SCAVENGING_FLUX: dust wet Deposition by Convective Precipitation Flux fluxes (kg/m2/sec)

SEASALT_EMISSION_FLUX: sea salt emission mass flux (kg/m2/sec)

SEASALT_SEDIMENTATION_FLUX: sea salt sedimentation mass flux (kg/m2/sec)

SEASALT_DRY_DEPOSITION_FLUX: sea salt dry deposition mass flux (kg/m2/sec)

SEASALT_WET_DEPOSITION_FLUX: sea salt wet deposition by large scale precipitation mass flux (kg/m2/sec)

SEASALT_SCAVENGING_FLUX: sea salt wet deposition by convective precipitation mass flux (kg/m2/sec)

BC_EMISSION_FLUX: black carbon emission mass flux (kg/m2/sec)

BC_SEDIMENTATION_FLUX: black carbon sedimentation mass flux (kg/m2/sec)

BC_DRY_DEPOSITION_FLUX: black carbon dry deposition mass flux (kg/m2/sec)

BC_WET_DEPOSITION_FLUX: black carbon wet deposition by large scale precipitation mass flux (kg/m2/sec)

BC_SCAVENGING_FLUX: black carbon wet deposition by convective precipitation mass flux (kg/m2/sec)

OC_EMISSION_FLUX: particulate organic carbon emission mass flux (kg/m2/sec)

OC_SEDIMENTATION_FLUX: particulate organic carbon sedimentation mass flux (kg/m2/sec)

OC_DRY_DEPOSITION_FLUX: particulate organic carbon dry deposition mass flux (kg/m2/sec)

OC_WET_DEPOSITION_FLUX: particulate organic carbon wet deposition by large scale precipitation mass flux (kg/m2/sec)

OC_SCAVENGING_FLUX: particulate organic carbon wet deposition by convective precipitation mass flux (kg/m2/sec)

2. ngac.tCCz.a2dfHHH.grib2, where HHH=00, 03, …, 120 and CC=00, 12:

Data fields are instantaneous on 1x1 degree lat/lon grid.

SEASALT1: sea salt bin1 (diameter: 0.06-0.2 micron) mixing ratio (kg/kg)

SEASALT2: sea salt bin2 (diameter: 0.2-1 micron) mixing ratio (kg/kg)

SEASALT3: sea salt bin3 (diameter: 1-3 micron) mixing ratio (kg/kg)

SEASALT4: sea salt bin4 (diameter: 3-10 micron) mixing ratio (kg/kg)

SEASALT5: sea salt bin5 (diameter: 10-20 micron) mixing ratio (kg/kg)

BC1: black carbon hydrophobic dry (median diameter: 0.0236 micron), mixing ratio (kg/kg)

BC2: black carbon hydrophilic dry (median diameter: 0.0236 micron), mixing ratio (kg/kg)

OC1: particulate organic carbon hydrophobic dry (median diameter: 0.0424 micron), mixing ratio (kg/kg)

OC2: particulate organic carbon hydrophilic dry (median diameter: 0.0424 micron), mixing ratio (kg/kg)

SO4: sulfate dry (median diameter: 0.139 micron), mixing ratio (kg/kg)

3. ngac.tCCz.aod_$NM.grib2, where NM=11p1um, 1p63um, 340nm, 440nm, 550nm, 660nm, 860nm & CC=00,12:

Total aerosol optical depth at specified wavelengths (11.1, 1.63, 0.34, 0.44, 0.55, 0.66, and 0.86 micron).  Please note:  NGACv1 total aerosol optical depth is from dust only. NGACv2 total aerosol optical depth is from multiple species including dust, sea salt, sulfate, black carbon and particulate organic carbon.

New fields:

ngac.t00z.aod_550nm.grib2 file also contains aerosol optical depth at 550nm from each species: dust, sea salt, sulfate, organic carbon and black carbon.

**Acknowledgement:**  The development of GBBEPx smoke emissions is funded by Joint Center for Satellite Data Assimilation (JCSDA) Science Development and Implementation (JDSI).  The NGACv2 tuning and evaluation work conducted by the UAlbany team (Lu, Chen, and Wei) has been partially supported by NOAA Modeling, Analysis, Prediction, and Projections (MAPP) Climate Test Bed (CTB) (award number NA14OAR4310182) and NOAA Next Generation Global Prediction System (NGGPS) Research-to-Operations Initiative (R2O) (award number NA15NWS4680008).  The authors thank Dr. R. Perez and Dr. S. Kivalov (University at Albany, State University of New York) for evaluating the impact of NGACv2 aerosol forecasts on the estimation of surface insolation, we also thank the Space Science and Engineering Center at University of Wisconsin-Madison for providing visible satellite images (http://www.ssec.wisc.edu/) and NASA Earth Observations for providing climatology data (https://earthobservatory.nasa.gov/GlobalMaps) in the surface insolation estimation experiment. The authors thank Dr. L.

Pan and Dr. H. Kim for evaluating NGACv2 impact on CMAQ forecast. The authors appreciate the multi-model ensemble work done by the NRL for ICAP. The lead author J. Wang gives special thanks to NCEP EMC management for project support and EMC colleagues for their scientific and technical inputs, including Shrinivas Moorthi, Yu-Tai Hou, Mark Iredell, and Boi Vuong. The lead author is grateful for the operational implementation support she got from NCEP NCO

colleagues: Simon Hsiao, Xiaoxue Wang, Steven Earle and Rebecca Cosgrove. The implementation evaluation by Craig Long, Peng Xian, and Jeff Reid are also greatly appreciated. Lastly, the authors would like to thank the two anonymous reviewers for their valuable suggestions and comments.

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

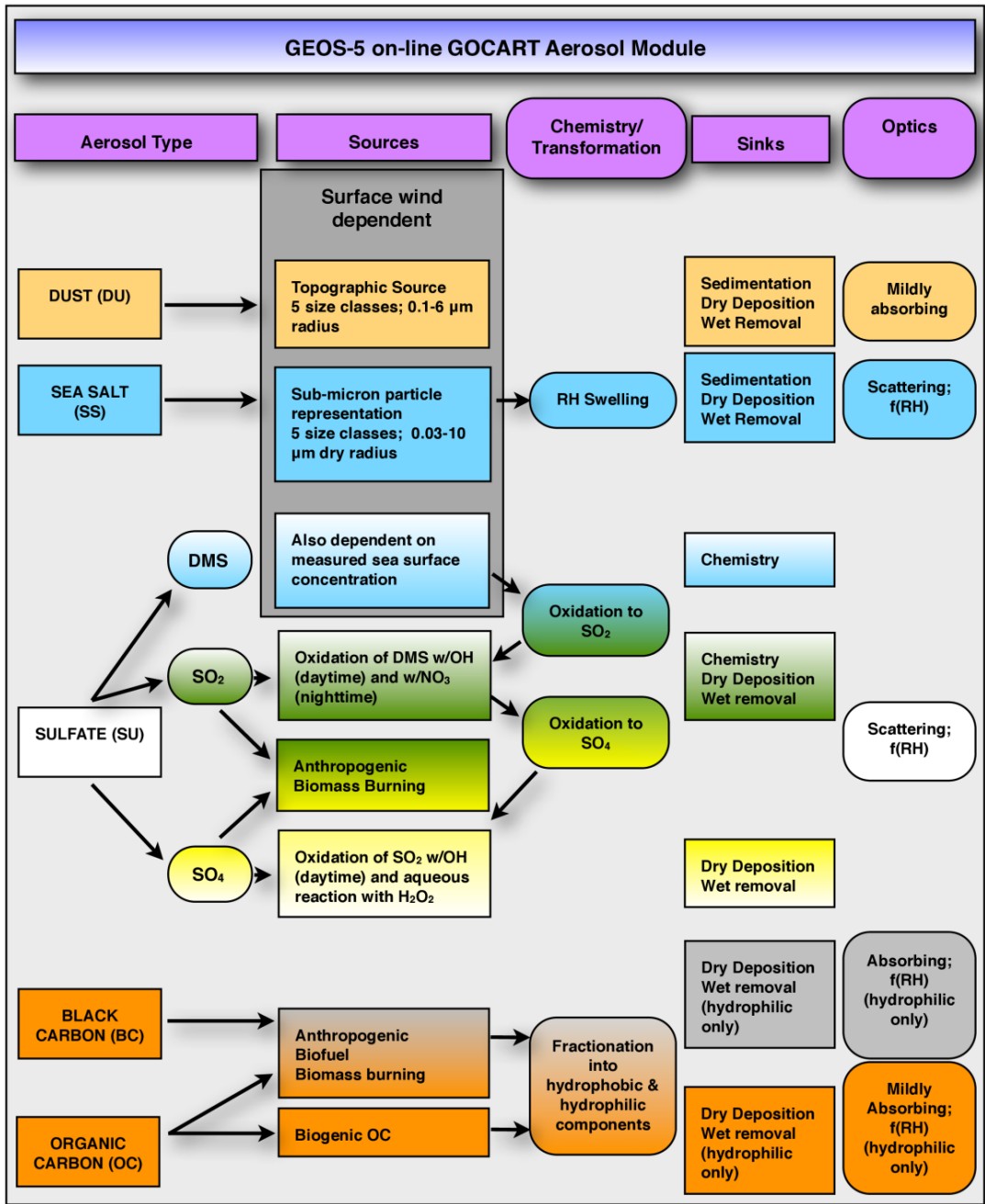

**Figure 1: Summary of aerosol modules in GEOS5 (Colarco et al. 2010). This aerosol module is adopted NGACv2. (Provided by P. Colarco, NASA/GSFC).**

**Table 1. Aerosol and precursor emissions in NGACv2.**

| Aerosol type | Sources | Temporal Resolution |
|---|---|---|
| Dust | Wind-driven emissions with Ginoux et al. (2001) static topographic depression map | Model |
| Sea Salt | Wind-driven emissions | Model |
| Biogenic terpene | Guenther et al (1995) | Monthly-mean climatology |
| Di-Methyl Sulfide (DMS) | Lana et al. (2011) | Monthly-mean climatology |
| Biomass Burning (SO2, OC, BC) | GBBEPx (Zhang et al., 2014) | Daily-varying |
| Anthropogenic SO2 | EDGAR V4.1 (European Commission, 2010) | Monthly-varying |
| Anthropogenic SO4, POM and BC | AeroCom Phase II (HCA0 v1, Diehl et al. 2012) | Annually-varying |
| International Ships SO2 | EDGAR V4.1 (European Commission, 2010) | Annually-varying |
| International Ships SO4, POM, and BC | AeroCom Phase II (HCA0 v1; Diehl et al. 2012) | Annually-varying |
| Aircraft SO2 | AeroCom Phase II (HCA0 v1; Diehl et al. 2012) | Monthly-varying |

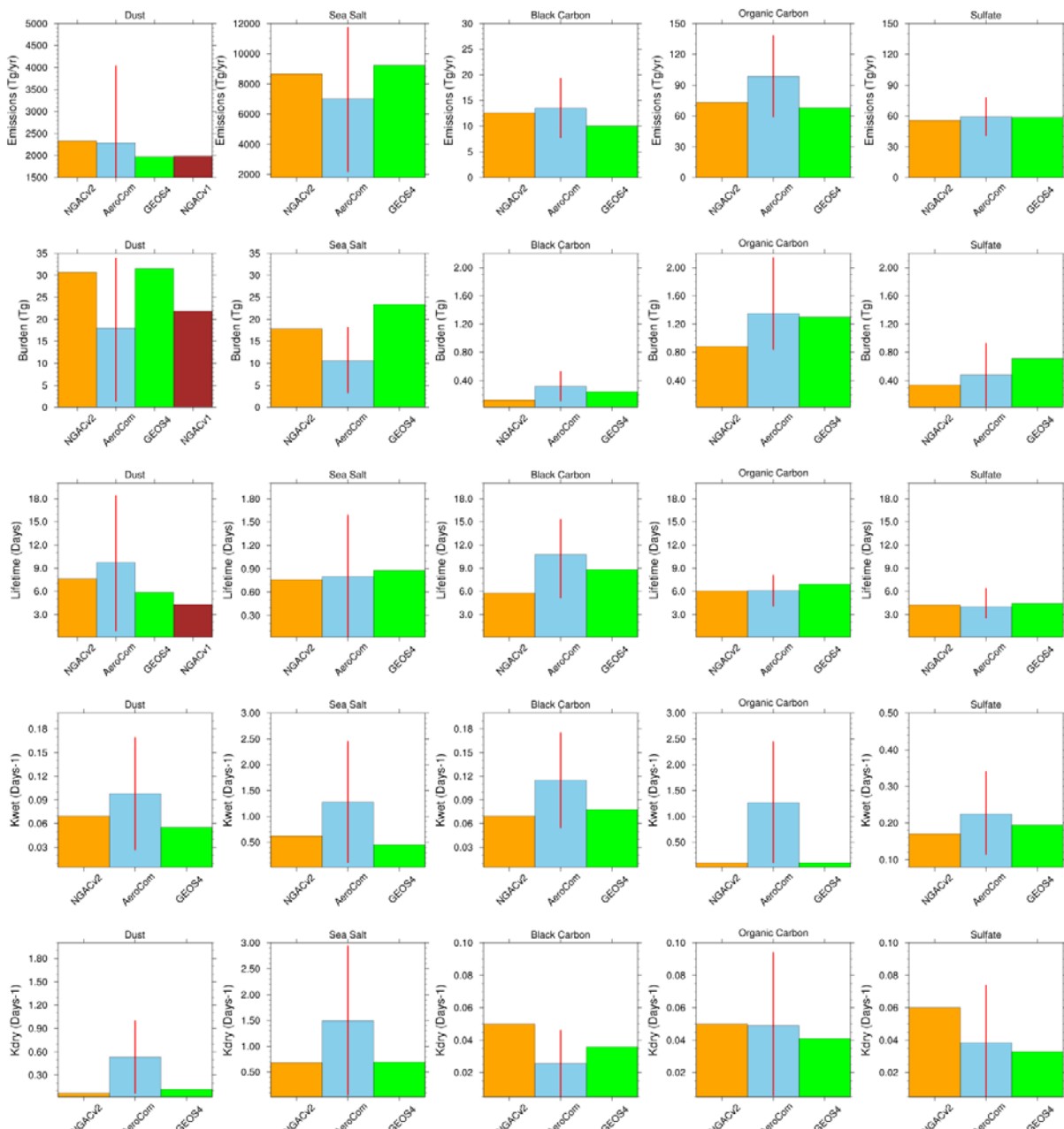

**Figure 2: Global annual total aerosol emissions and annual average aerosol burdens, lifetimes, and loss frequencies in NGACv2, AeroCom models, GOES4 and NGACv1 (dust only). For AeroCom models, the red vertical lines show the maximal and minimal values, and the bar shows the mean value. The first column is for dust, the second column is for sea salt aerosols, the third column is for black carbon, the fourth column is for organic carbon and the first column is for sulfate. Sulfate is for sulfur amount only. (Colarco. P. et al, J. Geophys. Res., 2010).**

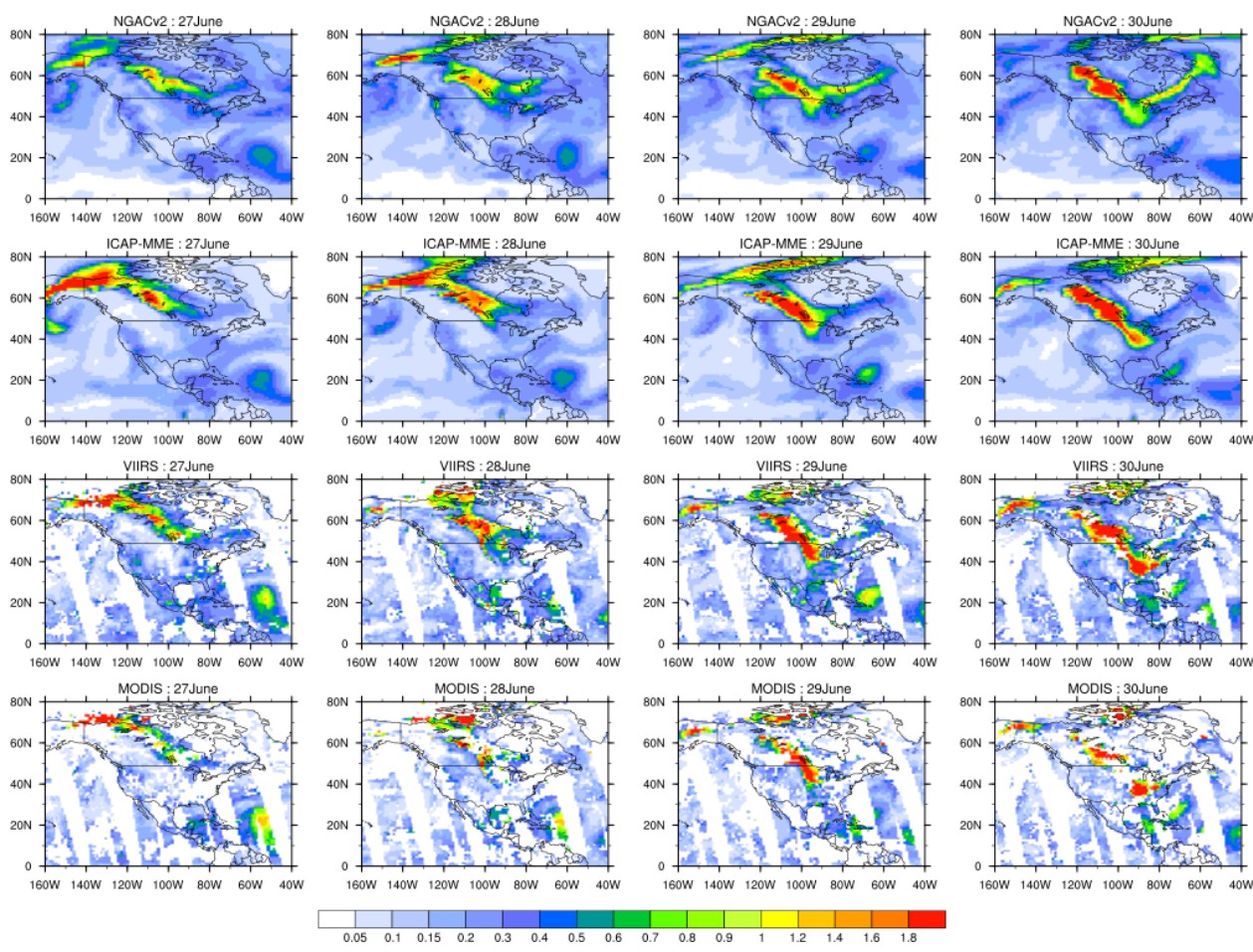

**Figure 3: June 27 to July 6 2015 smoke event from NGACv2 and ICAP forecasts and VIIRS and MODIS satellite observations.**

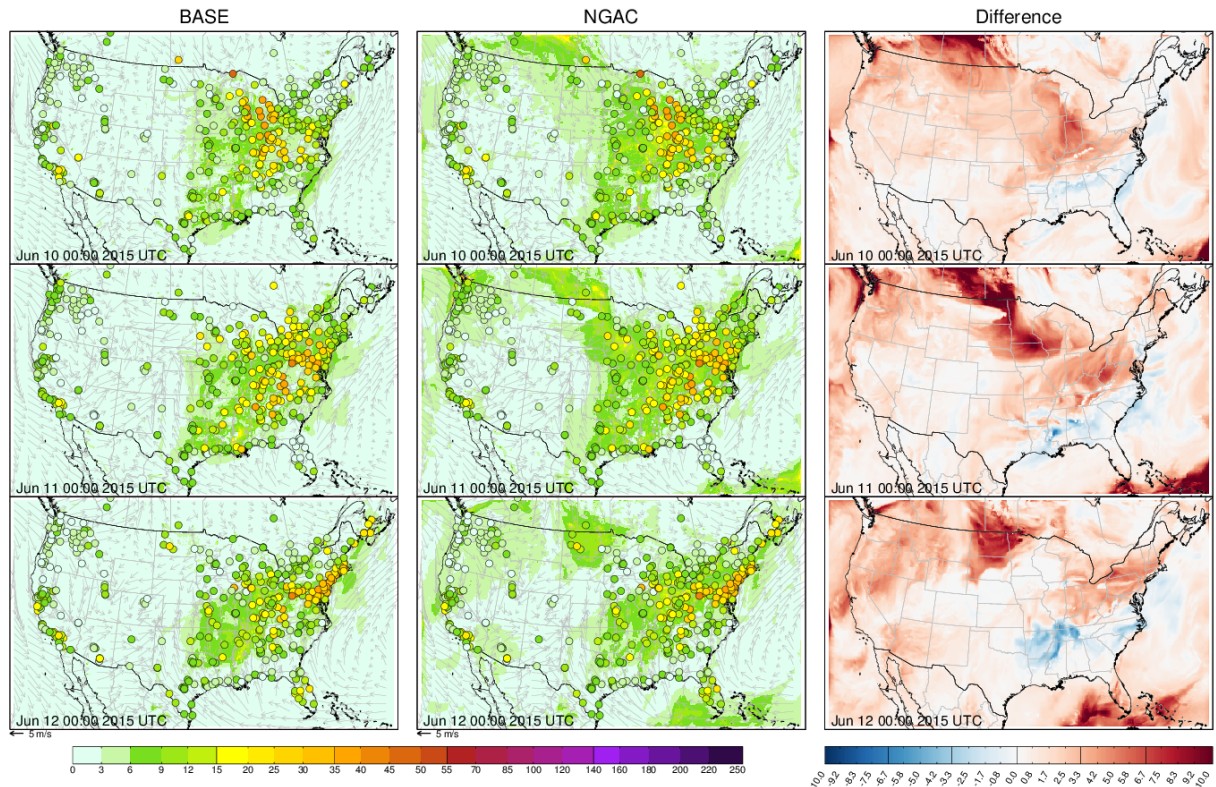

**Figure 4:** **PM2.5 forecasts from regional air quality model CMAQ during the smoke event on Jun 10-12, 2015. Base: using GEOS-Chem model 2006 monthly average as lateral boundary condition; NGAC: using NGAC forecast as lateral boundary condition, observations are the cycled colored dots. Differences between the two forecasts are shown in third column.**

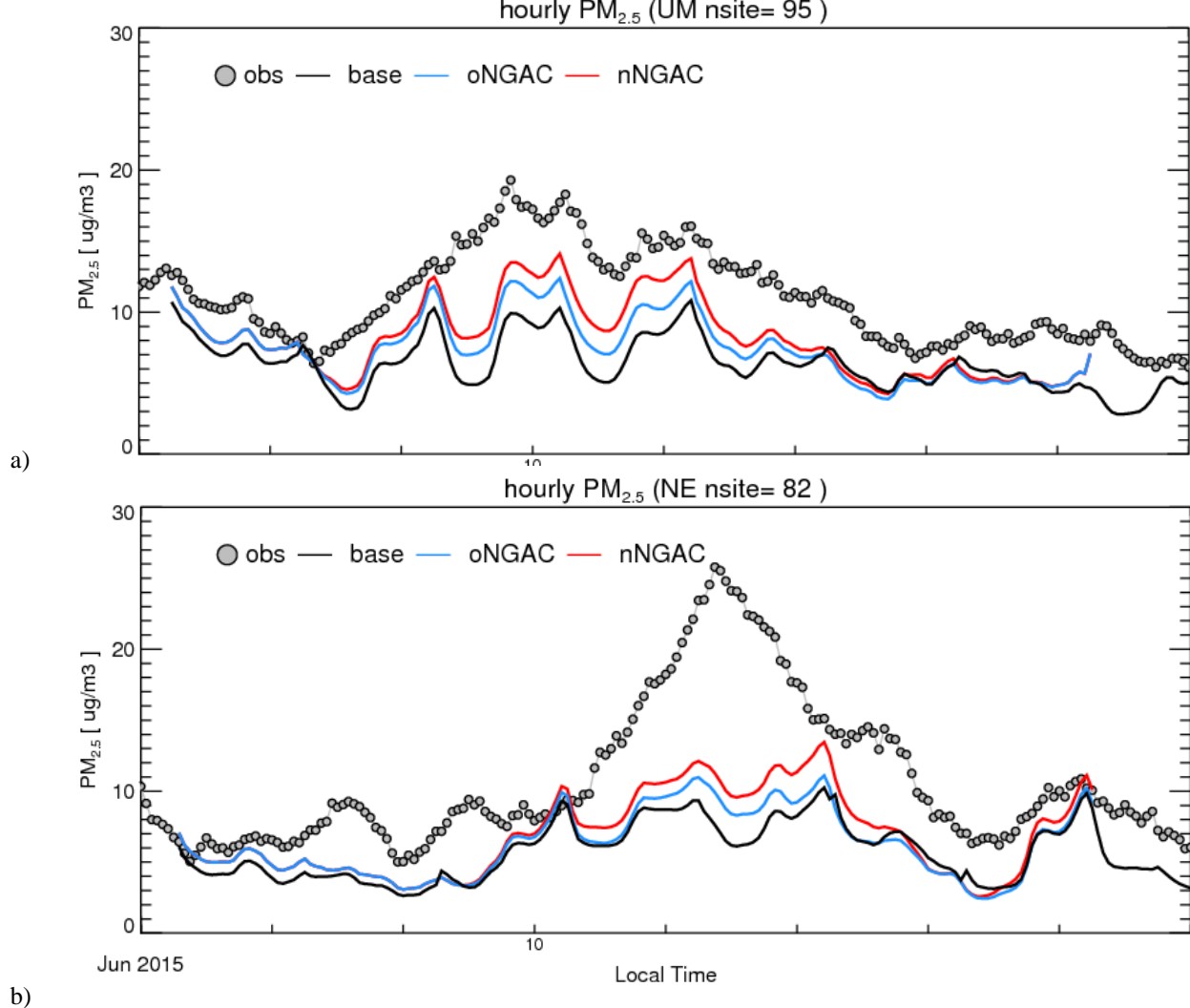

a)

b)

**Figure 5: Surface PM$_{2.5}$ with frontal passages in the June 9-12 2015 Canadian fire. a) is the averaged surface PM$_{2.5}$ from 95 sites in the middle of United States from June 7-15. b) is the averaged surface PM$_{2.5}$ from 82 sites in the northeast of United States. The line with circular dots is observations. The black line is the CMAQ forecast with climatology as lateral boundary condition. The blue line is the CMAQ using operational NGAC dust only forecast as lateral boundary condition. The red line is the CMAQ forecast using NGACv2 multi-species aerosol forecast.**

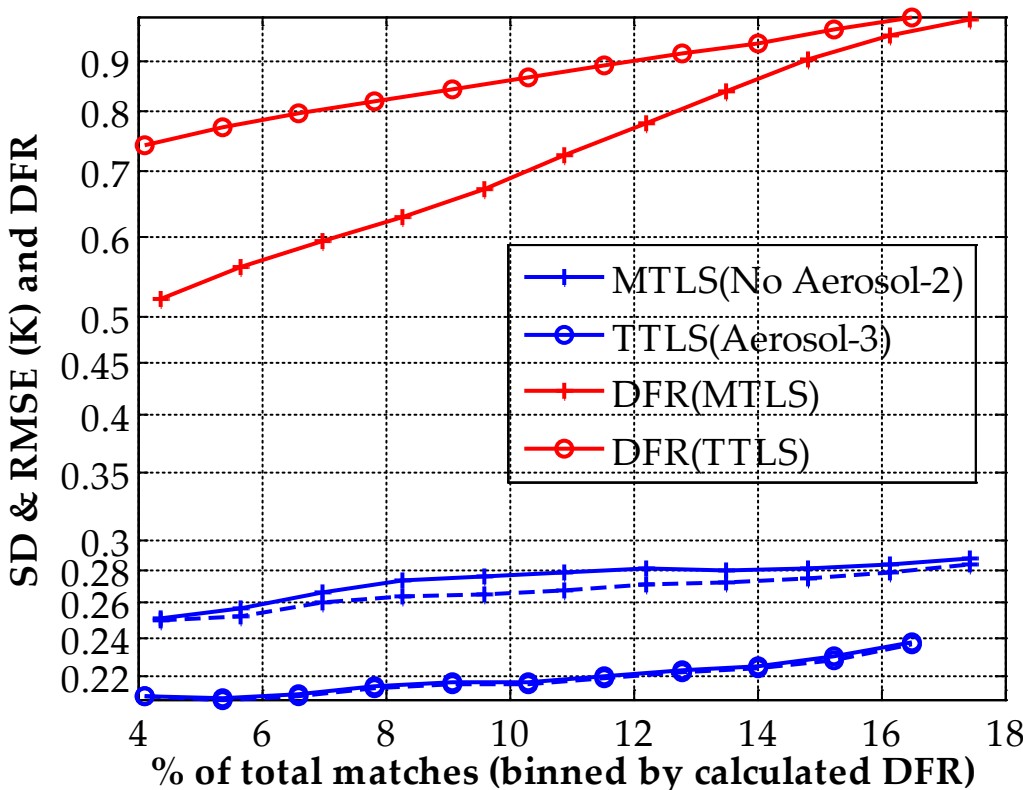

**Figure 6: Comparison of retrieval accuracy (blue lines) and algorithm sensitivity (Degrees of Freedom in Retrieval, red lines) of MTLS (plus) without aerosol and Truncated Total Least Squares (solid circles) using aerosol optical depth in the state vector for MODIS-Aqua data for January 2015. Evaluation is done against *i*Quam buoy data**

a)

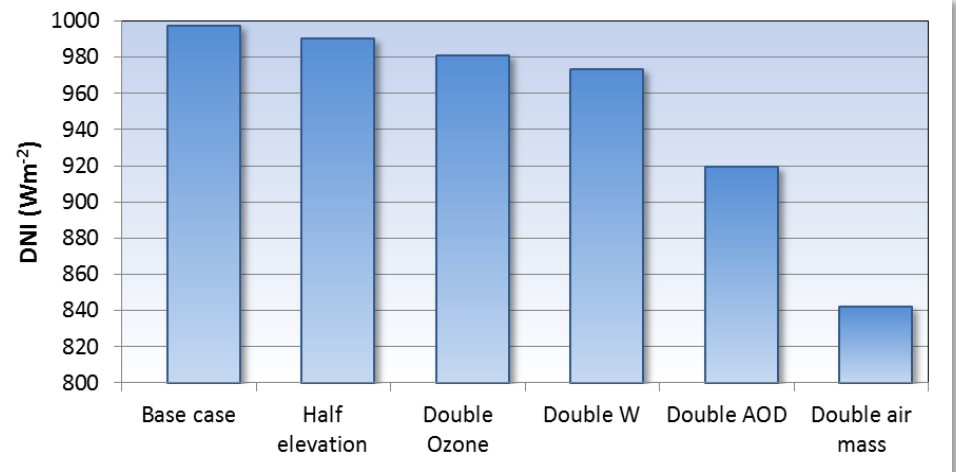

b)

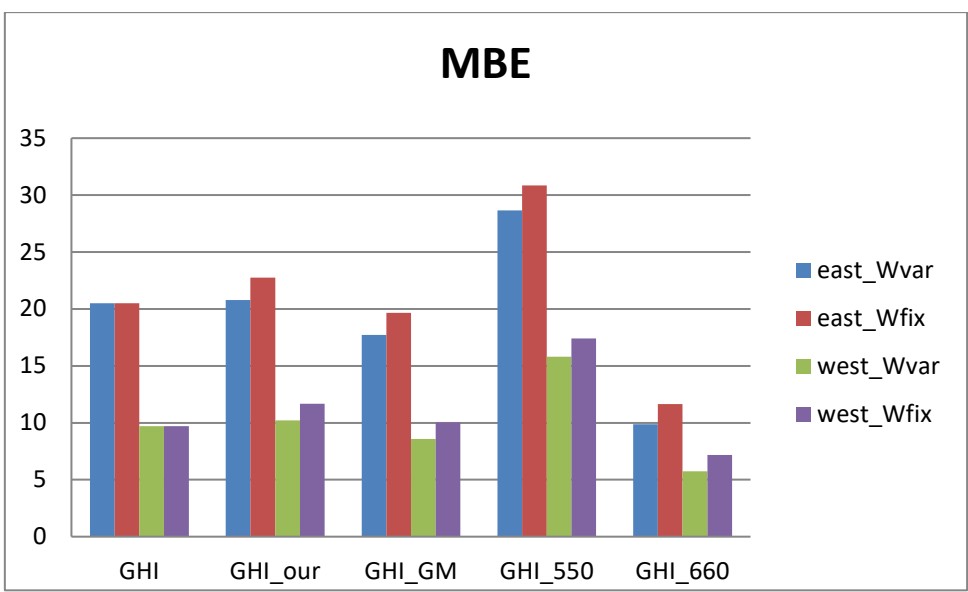

Figure 7: a) Aerosol impact on DNI from Perez model estimations; b) NGACv2 AOD impact on Perez model solar energy estimation mean bias error (MSE); east- GOES eastern USA satellite; west – GOES western USA; Wvar – variable water vapor (GFS model); Wfix monthly averaged water vapor (NASA) ; "our" – monthly averaged AOD used in ASRC (NASA); "GM" – Gueymard AOD; "550" and "660" - NGACv2 AOD.

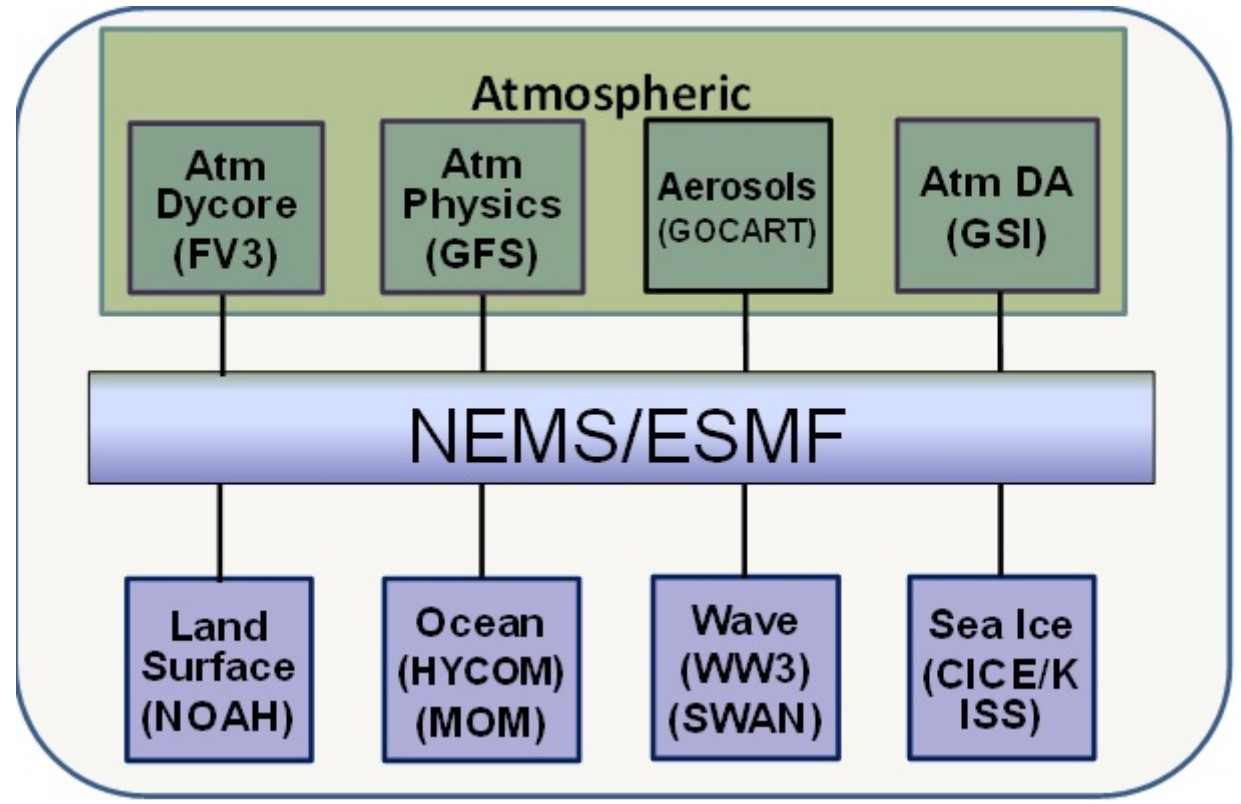

**Figure 8. Next Generation of Global Prediction System (NGGPS) implementation plan.**