# Peer review of "The implementation of NEMS GFS Aerosol Component (NGAC) Version 2.0 for global multispecies forecasting at NOAA/NCEP: Part I Model Descriptions"

_Geoscientific Model Development, 2017_

## Referee Comment (RC1) · Anonymous Referee #1 · 1 Feb 2018

**1   General comments**

This manuscript describes the implementation of the new version of the NEMS GFS Aerosol Component (NGAC) and shows some forecast results and impacts to some applications. The evaluations of the model are described in the companion paper.

The authors describe the aerosol forecast and its applications. Especially, a vegetation fire event case is demonstrated to examine its performance. The aerosol model

is based on GOCART aerosol module (Colarco et al. 2010): it is not very new or innovative but well documented and utilized by previous studies and suitable for operational forecast. While the manuscript is easy to read and the general performance of the NGACv2 model seems good, I think some of the specific points of the model and results need more description and revision: please see my Specific comments.

I recommend this manuscript to be published with a minor revision.

**2 Specific comments**

p.5, line 16- : "Sources for sulfate are ... biofuel and fossil fuel emissions from Aerosol Comparisons between Observations and Models (AeroCom) anthropogenic emissions." This contradicts with Table 1, which lists sources of anthropogenic SO2 is EDGAR V4.2 and International ships SO2 is EDGAR V4.1.

p.5, line 18: "DMS source uses climatology of oceanic DMS concentrations": How do you treat water-to-gas exchange (piston velocity) of DMS?

p.5, line 19-20: The biofuel and fossil fuel emissions are stated "climatology". Specifically, which years are taken to make the climatology and is it reasonable to use for current forecast?

p.8, line 26: Please expand the abbreviation ASRC at its first use.

p.7, Section 4.2: The forecasted AOD by NGACv2 show reasonable agreement with MODIS and VIIRS retrieved AOD and the multi-model ensemble forecast. However, NGACv2 does not show large AOD over the south of the Great Lakes that is shown in the satellite retrievals and the MME.

p.8, line 2: "The figure shows that using the NGAC forecast as the CMAQ lateral boundary condition significantly improved the CMAQ forecast": Figure 3 shows the impact of

providing lateral boundary from NGACv2 to CMAQ, but this does not necessary means improvements since it is not evaluated with observations.

p.8, line 4-: The inclusion of the lateral boundary from NGACv2 to CMAQ forecast does not seems to improve to reproduce the highest peak of PM2.5 in Fig. 4a and 4b.

p.8, Section 5.2 and 5.3: These results show sensitivities of the aerosol loadings to the SST retrieval and insolation on the Earth surface. However, these results do not guarantee the improvements of the real situations.

p.9, lines 7-12: It is strange that the conclusions of the validation by the companion paper (Bhattacharjee et al. 2017) is written in the conclusion of this manuscript.

**3   Technical corrections**

p.4 line 1: "Updates in NEMSGSM" -> "Updates in NEMS GSM"? (Whitespace between NEMS and GSM)

p.4, line 5: "(RRSM)" : Right parenthesis is in italic.

p.4, line 17: "NEMSGSM" -> "NEMS GSM"?

p.4, line 21: "other atmospheric aerosols" : Maybe "other" is not necessary

p.4, line 27 and hereafter in this paragraph: the parentheses of the citation are in brackets (AGU style?)

p.7, line 15: "imaginary" : "imagery" or just "image"?

p.7, line 12 and line 26: "Jun" -> "June" or "Jun."

p.8, line 8: "condtion" -> "condition"

p.8, line 29 and line 32: "Figure 6 a)" and "Figure 6 b)": Right parentheses are not

necessary.

p.8, line 30: What is the "Perez model"? (It is probably the "the semi-empirical GOES satellite global and direct horizontal irradiance estimation model developed at ASRC" but not stated.)

p.10 line 3: "inline": In other part of the manuscript, it is written as "in-line".

The model name: Mostly written as "NGACv2" but sometimes written as "NGAC v2" or "NGAC V2" or "NGAcv2": whitespace and capitalization are not consistent.

Please check the citation and reference list more carefully, What I found are:

p.2, line 21: Fan (2015) -> Fan et al. (2015)

p.3, line 2: Murphy (2014) -> Murphy et al. (2014)

P.3, line 5: Zhang (2016) -> Zhang et al. (2016)

p.4, line 9: (Han et AL. 2015) -> (Han et al. 2015)

p.5, line 22: (Gong S., 2003) -> (Gong 2003)

Table 1: Guenther et al. (1995) and Lana et al. (2011) are not found in the references.

Table 2: (Colarco. P. et al, Res., 2010) -> (Colarco. et al. 2010)?

References: Diehl et al. (2012) is ACP but the doi (url) there is acpd.

———————————————————

---

## Referee Comment (RC2) · Anonymous Referee #2 · 7 Feb 2018

This paper describes a substantial revision of NGAC, from a dust-only model to one including the usual fuller set of aerosol species (adding sea-salt, sulfate, black carbon and organic matter). Although there are no particularly novel scientific features compared to existing aerosol schemes, this paper serves to document the development of a well-used operational system. It is a well-written model description paper, and more extensive evaluation is provided in the companion paper (which is subject of a separate review). I would recommend publication in GMD subject to the following minor comments.

[Figure]

**p.2, lines 3–4:** "sea salt particles tend to reflect all the sunlight they encounter" – this is a rather simplistic description of their scattering behaviour.

**p.2, lines 10–12:** "Polluted air. . . leads to a weak hydrological cycle" – please clarify the limited conditions or scales for which this is true, since in general precipitation will be constrained by surface evaporation.

**p.2, lines 16–17:** I would suggest "aerosols **may** have significant impact" as the magnitude of such impacts outside idealised scenarios remains quite uncertain.

**p.3, line 25:** "full aerosols" would suggest that e.g. nitrate aerosol is included; perhaps "a wider range of aerosols"?

**p.5, line 14:** it would help to specify the actual resolution of this Gaussian grid rather than the spectral truncation, for those unfamiliar with the particular spectral–gridpoint mapping used.

**p.5, lines 17–18:** which AeroCom emissions dataset? There have been several recommendations for different phases of experiments – a specific link or reference would be helpful.

**p.5, line 21:** "Organic carbon has Terpene emission" – more detail on this rather terse statement would be welcome. Are terpenes emitted directly as organic carbon aerosol? Or emitted in the gas phase and subsequently converted to aerosol? From what inventory are these emissions derived, covering what sectors?

**p.6, lines 30–31:** "NGACv2 is closer to GEOS-4" – but does that mean it is closer to truth or observations?

**p.7, lines 26–29:** These plots compare results from CMAQ using NGACv2 vs GEOS-5 monthly boundary conditions, but observations should also be included to give some indication of which is performing better (as is done for the next example); otherwise the statement which follows that the forecast is improved is not justified.

**p.8, line 9:** although the run using NGACv2 is closer to observations here, it is worth noting that the values are still too low.

**p.8, line 14:** please define CRTM.

**p.8, lines 14–15:** please explain what is meant by the term "aerosol column density".

**p.8, line 23:** there appear to be two section 5.2s.

**p.8, line 32:** each of the different experiments presented here should be properly described.

**p.9, line 25:** should "GFSC" here be "GSFC" instead? Otherwise it should be defined.

**p.9, line 25–26:** the MAM aerosol scheme and MG cloud microphysics don't appear elsewhere in the paper – if their use is to be mentioned in the conclusions as more than a possibility for the future, more detail should be given at an appropriate point in the body of the paper.

**Table 2:** this is quite confusing with a lot of numbers, and I would consider finding a more accessible way to present the data (perhaps with the aid of a bar chart).

---

## Author Comment (AC1) · 17 Mar 2018

General comments:

This manuscript describes the implementation of the new version of the NEMS GFS Aerosol Component (NGAC) and shows some forecast results and impacts to some applications. The evaluations of the model are described in the companion paper. The authors describe the aerosol forecast and its applications. Especially, a vegetation fire event case is demonstrated to examine its performance. The aerosol model is based

on GOCART aerosol module (Colarco et al. 2010): it is not very new or innovative but well documented and utilized by previous studies and suitable for operational forecast. While the manuscript is easy to read and the general performance of the NGACv2 model seems good, I think some of the specific points of the model and results need more description and revision: please see my Specific comments. I recommend this manuscript to be published with a minor revision

- Response: The comments and suggestions from the referee #1 are greatly appreciated. All technical corrections have been made in the manuscript. Please see point-to-point response below for the specific comments.

Specific Comments:

p.5, line 16- : "Sources for sulfate are ... biofuel and fossil fuel emissions from Aerosol Comparisons between Observations and Models (AeroCom) anthropogenic emissions." This contradicts with Table 1, which lists sources of anthropogenic SO2 is EDGAR V4.2 and International ships SO2 is EDGAR V4.1.

- Response: We thank the reviewer #1 for pointing out the inconsistency between Table 1 and the text, the section 2.3 emissions has been updated to include sources for sulfur species and primary sulfate. Page 5, line 26 – page 6, line 3: "For sulfate aerosols, primary emissions of DMS, SO2, and SO4 are considered. Daily biomass burning emissions are taken from NESDIS GBBEPx dataset described above. Anthropogenic emissions of SO2 are taken from the Emissions Database for Global Atmospheric Research (EDGAR), version 4.1 (European Commissions, 2010). For anthropogenic emissions of primary sulfate, the AeroCom Phase II dataset (HCA0 v1, Diehl et al., 2012) is used. SO2 emissions from ocean-going ships are taken from EDGAR v4.1, and ship SO4 emissions, taken from AeroCom Phase II (HCA0 v1), are derived from gridded emissions data set of Eyring et al. (2005) using the EDGAR v4.1 SO2 ship emissions. Aircraft emissions of SO2 are derived from the AeroCom Phase II (HCA0 v1), which in turn is based on NASA's Atmospheric Effects of Aviation Program (AEAP) inventory. DMS emissions from marine algae are calculated from DMS concentrations and water-to-air transfer velocity (piston velocity). Monthly-varying DMS concentrations are taken from Lana et al. (2011). Piston velocity is computed from 2-m temperature and 10-meter wind following the empirical formula from Liss and Merlivat (1986). " We also made extensive revision on describing emissions sources of carbonaceous aerosols (page 6, lines 4-11) and sources of natural aerosols (page 6, lines 12-15).

p.5, line 18: "DMS source uses climatology of oceanic DMS concentrations": How do you treat water-to-gas exchange (piston velocity) of DMS?

- Response: Piston velocity is calculated from model temperature and wind following Liss and Merlivat (1986). Please see previous response on the corresponding manuscript revision.

p.5, line 19-20: The biofuel and fossil fuel emissions are stated "climatology". Specifically, which years are taken to make the climatology and is it reasonable to use for current forecast?

- Response: AeroCom emissions cover the period from 1979 to 2006. In NGACv2, we repeat the 2006 emissions. For real-time forecasts, the approach to use the latest available emission is widely used. In the manuscript, we add the following at section 2.3. Page6, lines 6-8: "For anthropogenic emissions, AeroCom Phase II data set (HCA0 v1) is used. This data set is based on gridded inventory from Bond et al. (2004) and yearly global emission trends compiled from Streets et al. (2008, 2009)"

p.8, line 26: Please expand the abbreviation ASRC at its first use.

- Response: suggested change is made in the manuscript. Page 10, line 4: Atmospheric Sciences Research Center (ASRC)

p.7, Section 4.2: The forecasted AOD by NGACv2 show reasonable agreement with MODIS and VIIRS retrieved AOD and the multi-model ensemble forecast. However, NGACv2 does not show large AOD over the south of the Great Lakes that is shown in

the satellite retrievals and the MME.

- Response: It is true that NGACv2 show reasonable agreement with satellite data and the MME, but it does not catch the large AOD over the south of the Great Lakes while MME does. Less satisfactory performance in NGAC v2 with respect to ICAP-MME suggest the need for additional model tuning. The authors believe increased resolution and aerosol data assimilation (DA) in the future NGAC implementation can also help to improve the NGACv2 performance. However it is not surprising that the multi-model ensemble mean from ICAP-MME outperforms a single model without DA. Manuscript is modified to provide the explanation (page 8, lines 13-23). Page 8, lines 13-23: "ICAP-MME total AOD products are generated from aerosol forecasts from four well-established aerosol models, including NASA/GSFC, ECMWF, Naval Research Laboratory (NRL), and Japan Meteorological Agency (JMA). Near-real-time satellite based smoke emissions are used by the four ICAP core models, e.g., QFED2 for NASA/GSFC, Fire Locating and Modeling of Burning Emissions (FLAMBE) by NRL, and Global Fire Assimilation System (GFAS) by ECMWF and JMA. In additional, aerosol data assimilation has been utilized by all these models to constrain the modelled AOD errors and bring modelled AODs closer to the satellite observations. Less satisfactory performance in NGAC v2 with respect to ICAP-MME suggest the need for additional model tuning. However, the performance differences cannot be attributed to NGACv2 model deficiency alone. Lynch et al. (2016) reported that model tuning process is equally as significant as data assimilation on the model performance. Sessions et al. (2015) reported that ICAP-MME out performs the participating members, providing a valuable aerosol forecast guidance. Therefore, the results that multi-model ensemble from four well-established models with data assimilation capabilities outperforms a single model without data assimilation is somehow anticipated."

p.8, line 2: "The figure shows that using the NGAC forecast as the CMAQ lateral boundary condition significantly improved the CMAQ forecast": Figure 3 shows the impact of providing lateral boundary from NGACv2 to CMAQ, but this does not necessary means

improvements since it is not evaluated with observations.

- Response: The Figure 4(Figure 3 before manuscript modification) is updated to show the impact of providing lateral boundary from NGACv2 to CMAQ forecast. Observations of PM2.5 (cycled dots) and synoptic condition (wind vector and pressure) are provided in the plots of CMAQ forecast with NGACv2 as lateral boundary condition (plots in the middle column). This figure shows that the CMAQ forecast with NGACv2 as lateral boundary condition matches observations better compared to BASE CMAQ forecast. The manuscript has been revised (page 9, lines 6-8). Page 9, lines 6-8 "The middle panel is the PM2.5 forecast from CMAQ during the same period using NGACv2 multi-species aerosols as the lateral boundary condition. PM2.5 observations (cycled dots) and synoptic condition (wind vector and pressure) are also shown in this panel to compare CMAQ forecast with observations."

p.8, line 4-: The inclusion of the lateral boundary from NGACv2 to CMAQ forecast does not seem to improve to reproduce the highest peak of PM2.5 in Fig. 4a and 4b.

- Response: while CMAQ forecast with NGACv2 dynamic LBC fails to capture highest peak in PM2.5, it produces the best agreement with observed PM among the three CMAQ experiments shown in Figure 5 (Figure 4 before manuscript modification). The manuscript has been revised. Page 9, line 20, adding "even though the peak of PM2.5 in this run is still lower than the observations."

p.8, Section 5.2 and 5.3: These results show sensitivities of the aerosol loadings to the SST retrieval and insolation on the Earth surface. However, these results do not guarantee the improvements of the real situations.

- Response: The authors agree that the sensitivities of the aerosol loadings do not guarantee improvement of the SST retrieval at this moment, but aerosol information provided by operational NGAC model makes quantitative evaluation possible of the aerosol impact on the SST retrieval and provides a path to develop statistical tools using the aerosol products to improve the SST retrieval. Section 5.2 is revised accordingly (page 10, lines 1-3). Page 10, lines 1-3: "The NGACv2 aerosol products make it possible to design the TTLS scheme; further development is required to improve the SST retrieval using aerosol products in real operation."

However section 5.3 does show that with NGACv2 AOD at 660nm the GHI mean bias error is significantly reduced compared to the GHI MSE in other experiments. The authors do expect potential improvement of GHI estimation in the real situations. Following Manuscript is revised (page 10, lines 22-23). Page 10, lines 22-23: "The results indicate potential improvement in the operational insolation estimate using NGAC AOD at 660nm."

p.9, lines 7-12: It is strange that the conclusions of the validation by the companion paper (Bhattacharjee et al. 2017) is written in the conclusion of this manuscript.

- Response: The authors would like to give general description on model, model performance and future work as a summary. Because future NGAC development work will be transitioned to the new FV3GFS based forecast system, the current model performance will be used as baseline for the transition. Following changes are made in the manuscript (page 10, lines 28-29 and page 11, lines 22-24). Page 10, lines 28-29: "Because the results will be used as baseline for some future development work described below, a general description of the NGACv2 evaluation is shown here." Page 11, lines 22-24: "NGAC global aerosol forecast capability is now being transitioned to the FV3GFS system; the NGACv2 forecast performance described above will be used as baseline to evaluate the FV3GFS based aerosol system."

---

## Author Comment (AC2) · 17 Mar 2018

General comments:

This paper describes a substantial revision of NGAC, from a dust-only model to one including the usual fuller set of aerosol species (adding sea-salt, sulfate, black carbon and organic matter). Although there are no particularly novel scientific features compared to existing aerosol schemes, this paper serves to document the development of a well-used operational system. It is a well-written model description paper, and more

extensive evaluation is provided in the companion paper (which is subject of a separate review). I would recommend publication in GMD subject to the following minor comments.

The comments and suggestions from the referee #2 are greatly appreciated. The specific comments have been addressed and point-to-point response is provided here

p.2, lines 3–4: "sea salt particles tend to reflect all the sunlight they encounter" – this is a rather simplistic description of their scattering behaviour.

- Response: The manuscript has been revised (page 2, lines 4-6). Page2, lines 4-6: "Sea salt particles scatter the incoming solar radiation and absorb the outgoing terrestrial radiation, with short and long wave radiation approximately the same order of magnitude, but in opposite sign (Lundgren, 2013)."

p.2, lines 10–12: "Polluted air. . . leads to a weak hydrological cycle" – please clarify the limited conditions or scales for which this is true, since in general precipitation will be constrained by surface evaporation.

- Response: it is true that the general precipitation is constrained by surface evaporation. However, the regional cooling aerosol radiation effect can result in lower evaporation, changing regional circulation and modifying the microphysics to reduce the rain. But there is uncertainty on general conclusion of aerosol impact on hydrological cycle. Manuscript is modified (page 2, lines 13-16). Page 2, lines 13-16: "Polluted air with an increased amount of aerosols tends to generate bright clouds reducing precipitation efficiently, which then leads to a weak regional hydrological cycle that affects the quality of fresh water over the tropics and the subtropics, especially in the Asian region which has the large tropical and subtropical aerosol emission sources (Ramanathan et al. 2001)."

p.2, lines 16–17: I would suggest "aerosols may have significant impact" as the magnitude of such impacts outside idealised scenarios remains quite uncertain.

- Response: Suggested change is made in the manuscript (page 2, line 19-20).

p.4, line 25: "full aerosols" would suggest that e.g. nitrate aerosol is included; perhaps "a wider range of aerosols"?

- Response: the suggested change is made in the manuscript. (page 5, line 31- page 6, line 1). Page 5, line 31- page 6, line 1: "In NGACv2 the GOCART module is updated and the suite of aerosol components is turned on to predict a wider range of aerosols."

p.5, line 14: it would help to specify the actual resolution of this Gaussian grid rather than the spectral truncation, for those unfamiliar with the particular spectral gridpoint mapping used.

- Response: The Gaussian grid resolution 100 km is added in the manuscript (Page 5, lines 23-24). Page 5, lines 23-24: "Emissions datasets are re-gridded to the native model grid (i.e., T126 Gaussian grid, about 100km horizontal resolution)."

p.5, lines 17–18: which AeroCom emissions dataset? There have been several recommendations for different phases of experiments – a specific link or reference would be helpful.

- Response: AeroCom PhaseII (HCA0 v1, Diehl T. 2012) emissions are used. Reference is added in the manuscript. Section 2.3 has been revised to provide additional information on emissions (Page 5, lines 28-29). Page 5, lines 28-29: "For anthropogenic emissions of primary sulfate, the AeroCom Phase II dataset (HCA0 v1, Diehl et al., 2012) is used."

p.5, line 21: "Organic carbon has Terpene emission" – more detail on this rather terse statement would be welcome. Are terpenes emitted directly as organic carbon aerosol? Or emitted in the gas phase and subsequently converted to aerosol? From what inventory are these emissions derived, covering what sectors?

- Response: Ten percent of the Terpene emission is converted to organic carbon aerosols through the oxidation of gas-phase precursors following Chin et al (2002).

The emission is from IGAC-GEIA 1990 inventory (Guenther et al. 1995); it includes isoprene, monoterpenes, other reactive VOC (ORVOC), and other VOC (OVOC) emissions. Manuscript is revised (page 6, lines 9-11). Page 6, lines 9-11: "Emissions of terpene from vegetation are oxidized to produce OC aerosols. Biogenic emissions are treated following Chin et al. (2002) using a monthly varying Global Emissions Inventories Activity (GEIA) inventory (Guenther et al. 10 1995)"

p.6, lines 30–31: "NGACv2 is closer to GEOS-4" – but does that mean it is closer to truth or observations?

- Response: this statement does not mean that NGACv2 is closer to truth or observations. We only compare NGACv2 with GEOS-4 and AeroCom model suite. Following explanation is added to manuscript (page 7, line 30 – page 8, line 2). Page 7, line 30 – page 8, line 2: "It is worth noting that the NGAC emissions, budget, and lifetime presented here are in the context of the AeroCom model suite. Evaluation of aerosol budget, emissions, and lifetime using observations is beyond the scope of this study."

p.7, lines 26–29: These plots compare results from CMAQ using NGACv2 vs GEOS-5 monthly boundary conditions, but observations should also be included to give some indication of which is performing better (as is done for the next example); otherwise the statement which follows that the forecast is improved is not justified.

- Response: The Figure 4 (Figure 3 before manuscript modification) is updated to show the impact of providing lateral boundary from NGACv2 to CMAQ forecast. Observations of PM2.5, (cycled dots) and synoptic condition (wind vector and pressure) are added in the plots of CMAQ forecast with NGACv2 as lateral boundary condition (plots in the middle column). This figure shows that the CMAQ forecast with NGACv2 as lateral boundary condition matches observations better compared to BASE CMAQ forecast. The manuscript has been revised (page 9, lines 6-8). page 9, lines 6-8: "The middle panel is the PM2.5 forecast from CMAQ during the same period using NGACv2 multi-species aerosols as the lateral boundary condition. PM2.5 observations (cycled

dots) and synoptic condition (wind vector and pressure) are also shown in this panel to compare CMAQ forecast with observations."

p.8, line 9: although the run using NGACv2 is closer to observations here, it is worth noting that the values are still too low.

- Response: The comment has been added in the manuscript (page 9, lines 19-20). Page 9, lines 19-20: "It is clear that the run with the NGACv2 forecast is closer to observations than the runs from the other experiments even though the peak of PM2.5 in this run is still lower than the observations"

p.8, line 14: please define CRTM.

- Response: The definition of CRTM is added in the manuscript as "Community Radiative Transfer Model" (page 9, line 25).

p.8, lines 14–15: please explain what is meant by the term "aerosol column density".

- Response: Because there is no direct derivation of aerosol retrieval in current SST retrieval algorithm, the three dimensional NGACv2 aerosol information is added in CRTM model, the Jacobian values for all aerosols and all altitude levels calculated from CRTM output are integrated as the aerosol retrieval. The aerosol column density of all aerosols is the single Jacobian value representing the derivative of radiation transfer equation with respect to a single variable. Manuscript is revised (page 9, lines 26-28). Page 9, lines 26-28: "Aerosol column density (ACD) of all aerosols is represented by the single Jacobian value that are calculated in CRTM representing the derivative of radiation transfer equation with respect to a single variable, the ACD is then included in the state vector for the MODIS-Aqua SST retrieval."

p.8, line 23: there appear to be two section 5.2s.

- Response: the section of "Insolation on the earth surface estimation" is changed to section 5.3.

p.8, line 32: each of the different experiments presented here should be properly described.

- Response: Manuscript is modified to add description of the experiments. Page 10, lines 14-23: "Figure 7b (Figure 6b before manuscript modification) shows aerosol impact on the mean bias error (MBE) of Global Horizontal Irradiance (GHI) from Perez Model. The GHI is computed from Perez Model with AOD and water vapor from different sources. Five sets of experiments were conducted to show the MBE. "GHI" is the control experiment with monthly averaged AOD from NASA Earth Observations (NEO) and water vapor from NEO or GFS. Validation are performed on two sets of sites, one set is validated against GOES eastern US satellite (GOES-13), the other against GOES western US satellite (GOES-15). "GHI_our" experiment is using the monthly averaged AOD from ASRC. "GHI GM" is using Gueymard AOD (Gueymard, 2008). GHI with NGACv2 AOD at 550nm and GHI with NGACv2 AOD at 660nm are also shown. It is found that GHI MBE is the smallest for the experiment using NGAC AOD at 660nm for the 2016 spring period. The results indicate potential improvement in the operational insolation estimate using NGAC AOD at 660nm."

p.9, line 25: should "GFSC" here be "GSFC" instead? Otherwise it should be defined.

- Response: The future work section has been revised to emphasize near term work of transitioning NGAC to FV3GFS based system. This sentence is removed.

p.9, line 25–26: The MAM aerosol scheme and MG cloud microphysics don't appear elsewhere in the paper – if their use is to be mentioned in the conclusions as more than a possibility for the future, more detail should be given at an appropriate point in the body of the paper.

- Response: There is some uncertainty on the time line of implementing the MAM aerosol schemes and MG cloud microphysics due to the transition to FV3 based atmospheric system. The two lines are removed.

Table 2: This is quite confusing with a lot of numbers, and I would consider finding a more accessible way to present the data (perhaps with the aid of a bar chart)

- Response: Table 2 is presented as a bar chart (Figure 2, see attached figure); the manuscript is revised (page 21). Following is the caption for Figure 2: Figure 2: Global annual total aerosol emissions and annual average aerosol burdens, lifetimes, and loss frequencies in NGACv2, AeroCom models, GOES4 and NGACv1 (dust only). For AeroCom models, the red vertical lines show the maximal and minimal values, and the bar shows the mean value. The first column is for dust, the second column is for sea salt aerosols, the third column is for black carbon, the fourth column is for organic carbon and the first column is for sulfate. Sulfate is for sulfur amount only. 5 (Colarco. P. et al, J. Geophys. Res., 2010).
* * *
[Figure]

**Fig. 1.** Figure 2 (previous table 1)

---

## Author Response (AR2)

**The implementation of NEMS GFS Aerosol Component (NGAC) Version 2.0 for global multispecies forecasting at NOAA/NCEP: Part I Model Descriptions**
**By Jun Wang et al.**

Response to the final review comments.

General comments:

*The authors have appropriately addressed all the comments raised during the discussion phase. The additional Figure 2, and inclusion of observations in Figure 4 (was 3) are greatly appreciated.*

*There's one technical recommendation I would make before publication, which is that in Figure 4 the observations should be superimposed on both models (BASE and NGAC) rather than only the newer one, for easier comparison of how well they perform.*

*Aside from that, I am pleased to recommend the revised version for publication in ACP.*

- Response: The authors really appreciate the reviewer's comments and suggestions. Figure 4 is updated with observations superimposed on both experiments BASE and NGAC for easy comparison. The synoptic features are removed from the second NGAC plots as we focus on to compare PM2.5 forecast and there is no synoptic condition discussion in the context. The section 5.2 is updated and the updated figure is attached.

  Page 9, line 4 – line 12:

  Figure 4 shows an event on June 10-12, 2015 when smoke from Canada was moving into the United States. The left side panel is the PM2.5 forecast on June 10th, 11th and 12th from the CMAQ run using GEOS-Chem model 2006 monthly average values for all the aerosol species at the lateral boundary. The middle panel is the PM2.5 forecast from CMAQ during the same period using NGACv2 multi-species aerosols as the lateral boundary condition. PM2.5 observations in cycled dots are also shown in both panels to compare CMAQ forecast with observations. The right panel is the difference between the two runs. The figure shows that no smoke was predicted over central Canada and the US in the run using the climatology as the lateral boundary condition; while the run using NGAC multi-species aerosols as the boundary condition shows a large amount of smoke passing the US-Canadian border and coming across the Great Lakes region. The figure shows that using the NGAC forecast as the CMAQ lateral boundary condition significantly improved the CMAQ forecast.

[Figure]

**Figure 4: PM₂.₅ forecasts from regional air quality model CMAQ during the smoke event on Jun 10-12, 2015. Base: using GEOS-Chem model 2006 monthly average as lateral boundary condition; NGAC: using NGAC forecast as lateral boundary condition, observations are the cycled colored dots. Differences between the two forecasts are shown in third column.**